# Effects of Post-Weld Heat Treatment on Microstructure and Mechanical Properties of the Brazed Joint of a Novel Fourth-Generation Nickel-Based Single Crystal Superalloy

**DOI:** 10.3390/ma16083008

**Published:** 2023-04-11

**Authors:** Zhipeng Zhang, Jide Liu, Chongwei Zhu, Yuyu Huang, Xinguang Wang, Yizhou Zhou, Jianjun Wang, Jinguo Li

**Affiliations:** 1Key Laboratory for Anisotropy and Texture of Materials (MOE), School of Materials Science and Engineering, Northeastern University, Shenyang 110819, China; 2Research Center for Metallic Wires, Northeastern University, Shenyang 110819, China; 3Shi-changxu Innovation Center for Advanced Materials, Institute of Metal Research, Chinese Academy of Sciences, Shenyang 110016, China

**Keywords:** fourth-generation nickel-based single crystal superalloy, brazing, post-weld heat treatment, mechanical properties

## Abstract

A novel fourth-generation nickel-based single crystal superalloy was brazed with Co-based filler alloy. The effects of post-weld heat treatment (PWHT) on the microstructure and mechanical properties of brazed joints were investigated. The experimental and CALPHAD simulation results show that the non-isothermal solidification zone was composed of M_3_B_2_, MB-type boride and MC carbide, and the isothermal solidification zone was composed of γ and γ’ phases. After the PWHT, the distribution of borides and the morphology of the γ’ phase were changed. The change of the γ’ phase was mainly attributed to the effect of borides on the diffusion behavior of Al and Ta atoms. In the process of PWHT, stress concentration leads to the nucleation and growth of grains during recrystallization, thus forming high angle grain boundaries in the joint. The microhardness was slightly increased compared to the joint before PWHT. The relationship between microstructure and microhardness during the PWHT of the joint was discussed. In addition, the tensile strength and stress fracture life of the joints were significantly increased after the PWHT. The reasons for the improved mechanical properties of the joints were analyzed and the fracture mechanism of the joints was elucidated. These research results can provide important guidance for the brazing work of fourth-generation nickel-based single crystal superalloy.

## 1. Introduction

With the continuous innovation and development of the aviation industry, the superalloy materials that have been widely used in aircraft engines are also improving [1,2,3]. From equiaxed crystal to columnar crystal, and finally to single crystal, the transverse and longitudinal grain boundaries (GBs) are eliminated, and the service temperature and strength of the alloy are greatly improved [4,5]. Nickel-based single crystal superalloy has a more stable structure because of its disordered γ matrix and γ′ phase with L1_2_-ordered structure. Therefore, it is the best candidate material for aviation engine turbine blades [6,7,8]. With the addition of Re and Ru elements, the fourth-generation nickel-based single crystal superalloy is designed to further improve the high temperature mechanical properties of the alloy [9,10,11].

Turbine blades, guide vanes and other engine hot end components in aircraft engines have very complex structures, which are difficult to complete by the traditional casting process [12,13]. In addition, various defects will inevitably be formed in the process of long-term service [14]. Therefore, it is essential to use welding technology to connect and repair them. Vacuum brazing technology overcomes the shortcomings of other welding technologies and has been proven to be one of the most ideal welding technologies in practical engineering applications [15,16,17,18]. Generally, the brazing process can be divided into three stages: filler alloy melting stage, isothermal solidification stage and post-weld heat treatment (PWHT) stage. Wang et al. [19] realized the joining of nickel-based single crystal superalloy by brazing technology and the tensile strength of the brazed joint could be close to the base metal (BM) after the PWHT. The isothermal solidification stage was focused on, but the microstructure evolution during PWHT was neglected. Cao et al. [20] found that the second phase in the joint can effectively improve the tensile strength of the IN718 alloy joint by adjusting different PWHT process parameters. Pouranvari et al. [21] eliminated boride in IN718 alloy joints by adjusting different PWHT process parameters. Previous studies have paid more attention to the PWHT of polycrystalline nickel-based superalloy joints [22,23,24,25]. Unfortunately, there is still a lack of study on the PWHT of high generation nickel-based single crystal superalloy brazed joints.

The brazing of the fourth-generation nickel-based single crystal superalloy has been preliminarily explored, but the PWHT of the brazed joints in this alloy system has not been studied. In this paper, the fourth-generation nickel-based single crystal superalloy was vacuum brazed with a Co-based filler alloy. The microstructure and mechanical properties of as-brazed and heat-treated joints were studied. The formation of precipitates in the joints and the microstructure evolution of the joints during heat treatment were discussed.

## 2. Experimental Procedures

A new type of fourth-generation nickel-based single crystal superalloy used in this experiment was based on solution strengthening and γ′-strengthening, and its chemical composition was listed in Table 1. The Co38CrNiWSiB filler alloy was prepared by vacuum induction melting gas atomization (VIGA) method, and then 100–200 mesh alloy powders were selected. The solidus and liquidus temperatures of the filler alloy are 1159 °C and 1208 °C, respectively. It has been proved that the brazing alloy can be used for brazing the fourth-generation nickel-based single crystal superalloy [26]. The chemical composition of the filler alloy was listed in Table 2.

The casting rod (Φ16 × 210 mm) of (001) oriented single crystal superalloy was prepared by the high-rate solidification (HRS) method. The microstructure of the single crystal rod was shown in Figure 1. The single crystal rod as the base metal after solution treatment, and the following two aspects were considered: (1) Elimination of dendrite segregation in as-cast single crystal rod. (2) The melting point of the filling alloy is far lower than the solution treatment temperature. The brazing experiments were arranged after the solid solution treatment, which prevented remelting of the filler alloy in the weld. Firstly, the base metal was cut with a wire electrical discharge machining (EDM), and the samples of microstructure and mechanical properties were cut into Φ16 × 3 mm and Φ16 × 35 mm, respectively. Second, the cut samples were polished smooth with 200 and 400 mesh sandpaper and cleaned with acetone for 15 min. Finally, the water-soluble adhesive was mixed with solder powder to form a paste, which was filled around the weld. It was worth mentioning that 100 µm nickel wire was used to control the weld gap. The assembled samples were heated in a vacuum brazing furnace. After the filler alloy melts, it was wetted and spread into the weld center under the capillary action. The brazed joint will be formed when the samples were cooled to room temperature in the furnace, and the vacuum degree was always kept below 4 × 10^−3^ Pa during this process. In addition, the brazing temperature was set between the liquidus temperature of the filler alloy (1208 °C) and the solution treatment temperature of the BM (1325 °C). In this paper, the same heat treatment process as the BM was used to carry out PWHT on the brazed joint. The brazing and heat treatment processes were shown in Figure 2. In order to show the joint status more succinctly, the brazed joint was defined as HT1, and the joint after PWHT was defined as HT2.

The microstructure of brazed joints was analyzed by scanning electron microscopy (SEM). The SEM samples were operated by standard metallographic procedures, and then the samples were etched with copper sulfate corrosion solution (20 g CuSO_4_ + 100 mL HCl + 5 mL H_2_SO_4_ + 80 mL H_2_O) for 3–5 s. The element concentration and distribution of the joint were analyzed by energy dispersive spectroscopy (EDS) and electron probe micro-analyzer (EPMA) techniques. Thermal Calc 2021a software using the TCNI10 database was used to calculate the residual liquid phase components in the joint. The phase distribution and grain boundaries in the joints were obtained by electron back-scattered diffraction (EBSD) and the data were analyzed by the channel 5 software package. The cross-section of the brazed joint was subjected to the Vickers hardness test with a 100 g load for 15 s. Select 3 points in each area to test the microhardness, and then take the average value. In addition, the high-temperature tensile (980 °C) and stress rupture tests (980 °C/60 MPa) were carried out on the joints under different conditions.

## 3. Results and Discussion

### 3.1. Microstructure of Brazed Joint

Figure 3a shows the overview microstructure of the brazed joint at 1260 °C for 90 min. It is observed that the brazed joint was composed of two different zones: the non-isothermal solidification zone (NSZ) and the isothermal solidification zone (ISZ). Due to insufficient brazing time, a large number of eutectic compound precipitates were formed in the NSZ. On the other hand, there is no diffusion affected zone (DAZ) near the BM in the brazed joint, which is obviously very beneficial to the properties of the BM. By amplification of the ISZ, it is found that the γ’ phase near the NSZ side is very small and at the nanometer level, while the γ’ phase on the side near the BM is relatively large. In order to better understand the formation of the precipitated phases in different regions conveniently, these two regions are redefined as ISZ-1 and ISZ-2, respectively, while the two precipitated phases are called secondary γ’ (S-γ’) and primary γ’ (P-γ’) phase, respectively, as shown in Figure 3c,d. The γ’ phase size in ISZ-2 is still quite different from that in the BM. The yellow dotted line can be used to distinguish ISZ-2 from BM successfully (Figure 3b).

By magnifying the NSZ, it is obvious that there are three different types of precipitates. The white skeleton precipitates occupy most of the area in the NSZ, and the volume fraction of gray strip precipitates and bright white block precipitates is relatively small, and the remaining area is composed of γ matrix, as shown in Figure 3e. In order to determine the element content of different precipitates, EDS technology was used to analyze different positions, and the results were listed in Table 3. It is found that the bright white block precipitates are rich in Ta, the gray strip precipitates are Cr-rich, and the white skeleton precipitates are rich in W and Ta. Since EDS cannot accurately detect light elements such as B and C, the EPMA technique was used for further qualitative analysis of the different precipitates, as shown in Figure 4. The results show that the bright white block precipitates are Ta-rich carbides, and the common forming elements of MC carbide are Ta, Ti, and Hf, so it can be inferred that this phase is MC carbide [27]. It is obvious that the gray strip precipitate is a Cr-rich boride. The white skeleton precipitate can be determined as W and Ta-rich boride. In addition, it is found that the Si element is uniformly distributed in the γ solid solution and do not form a silicide with other elements. This is mainly due to the low content of Si in the filler alloy and the high solubility of Si in the γ solid solution [28].

In this experiment, due to insufficient brazing time, the joint did not complete isothermal solidification. During the cooling process, the residual liquid phase gradually solidified and formed different precipitates. It can be seen from the above analysis results that the joint was mainly composed of boride, carbide, γ and γ’ phase composition. In order to better understand the formation of precipitates in the joint, the thermal-calc 2021a software was used to calculate the phase composition of the joint. Firstly, the solidification process of filler alloy without element diffusion was calculated. It can be seen from Figure 5a that a variety of precipitates are formed during the solidification of the filler alloy. However, the element diffusion between filler alloy and BM cannot be ignored in the actual brazing process. Therefore, the concentration of each element in the center of the joint after brazing can be calculated based on the simple law of conservation of mass, which can be represented by the following equation [29]: (1)$$C_{jab}^{i}=DC_{BM}^{i}+(1-D)C_{fa}^{i}$$
where $C_{jab}^{i}$ is the concentration of element i in the joint after brazing; $C_{BM}^{i}$ is the concentration of element i in the BM; $C_{fa}^{i}$ is the concentration of element i in the filler alloy; D represents the dissolution ratio, which can be expressed by the following formula:(2)$$D=\frac{M_{a}-M_{b}}{M_{a}}$$
where $M_{a}$ is the weight of the filler alloy in the joint after brazing; $M_{b}$ is the weight of the filler alloy before brazing. Since both the filler alloy and the BM are superalloys, it can be assumed that they have the same density. Therefore, the dissolution ratio D can also be represented by the following equation [30]:(3)$$D=\frac{M_{\max}-M_{0}}{M_{\max}}$$
where $W_{\max}$ is the maximum width of the joint after brazing; $W_{0}$ is the initial width of the brazed joint. Therefore, the dissolution ratio D can be brought into Equation (1), and the following equation can be obtained:(4)$$C_{jab}=\frac{M_{\max}-M_{0}}{M_{\max}}C_{BM}^{i}+(1-\frac{M_{\max}-M_{0}}{M_{\max}})C_{fa}^{i}$$

According to this equation, the concentration of each element in the joint can be calculated, and then the phase composition during solidification can be calculated using the thermal-calc software. The calculated phase composition results are illustrated in Figure 5b,c. The precipitation of the γ’ phase from the γ matrix indicates the diffusion of Al from the BM into the joint to combine with Ni to form the γ’ phase. Combined with the experimental results, the two borides can be tentatively identified as M_3_B_2_- and MB-type borides, respectively. The simulation results show the formation of M_23_C_6_, and the MC in the experimental results may be its intermediate transition phase, which is not transformed into M_23_C_6_. Since the experimental results are the result of the coupling of many factors, some errors between the actual solidification process and the theoretical solidification process are acceptable.

### 3.2. Effect of PWHT on the Microstructure of Brazed Joint

Figure 6 shows the microstructure of the brazed joint with PWHT. Compared with the joints before heat treatment (Figure 3), it can be clearly found that borides in NSZ change from large dense skeleton to discrete block distribution. According to the EBSD phase distribution map, the precipitated phases in the joint were identified as M_3_B_2_, MB and MC (Figure 7). This indicates that the simulation results have a certain accuracy. It also shows that the PWHT only changes the precipitation phase distribution, not the precipitation phase type. In addition, it can be seen from Figure 6b–d that obvious traces of interface connection were observed between ISZ-1 and ISZ-2 and between ISZ-2 and BM. The S-γ’ phase in ISZ-1 remained unchanged and remained at the nanometer level (Figure 6e). However, many needle-like compounds phases were precipitated from ISZ-1, and The EDS results indicate that the main chemical composition of the precipitated phase is Cr-5.6, Co-17.8, NI-11.3, Ta-15.6, W-42.4, Re-7.6 (wt%), which indicated that the precipitated phase is the same boride as the skeleton-like precipitated phase. It is observed that the P-γ’ phase in ISZ-2 changed from an irregular shape to a regular cubic shape and the size of the P-γ’ phase increased (Figure 6c). Meanwhile, the S-γ’ phase is observed in the γ matrix channel (Figure 6f). Furthermore, it can be seen from Figure 6g that the rafting of the γ’ phase and the coarsening of the γ matrix in the BM.

In order to better understand the formation of joints and the evolution of microstructure during PWHT, a series of simple schematic diagrams were drawn (Figure 8). In the first stage, the filler alloy completely melts when heated to the brazing temperature, and then wets and spreads on the surface of the BM under capillary action (Figure 8a). Under the action of concentration gradient, the filler alloy and the BM conduct the element mutual diffusion and begin to enter the isothermal solidification stage. At the same time, the B atoms combine with Ta, W, Cr and other atoms to form boride, thereby reducing the system energy. Immediately after that, the γ’ phase gradually precipitates in the γ solid solution and cools to room temperature to form a stable joint, as shown in Figure 8b. Finally, the brazed joint is subjected to PWHT, which is the focus of the work in this paper.

After long-term aging treatment, the boride volume fraction in NSZ decreased significantly. The M_3_B_2_-type boride dissolution temperature is found by thermal-calc software simulations to be upwards of 1300 °C (Figure 5c), which is significantly higher than the high-temperature aging treatment in this experiment, indicating that the boride elimination is controlled by interatomic solid-state diffusion. Pouranvari [21] reported in detail the evolution mechanism of boride elimination in the joint after PWHT, which mainly experienced two stages of precipitate break-up and dissolution. The dissolution temperature of MB-type boride (1048 °C) is lower than the high-temperature aging temperature (1150 °C), indicating that MB-type boride is dissolved during high-temperature aging. In addition, the volume fraction of MB-type boride in the joint is low, and it can be completely dissolved at high temperatures for a long time.

After PWHT, the S-γ’ phase in ISZ-1 changed little and the P-γ’ phase changed obviously in ISZ-2, this can be explained by the following discussion of this phenomenon. During high-temperature aging treatment, B atoms in NSZ diffuse to the surrounding. Due to the extremely low solubility of B in the Ni-based alloy, excessive B atoms will diffuse into ISZ-1 and combine with surrounding W and Ta atoms to form needle-like borides (Figure 8c).

It is well known that Ta is an important element in the formation of γ’ phase [31]. The lack of Ta will directly affect the precipitation of the γ’ phase. Moreover, the formation of needle-like borides repels the diffusion of Al elements into ISZ-1. Therefore, the lack of Ta and Al elements is the direct reason for the absence of significant changes in the S-γ’ phase in ISZ-1. In addition, the formation of needle-like boride leads to the hindrance of Al atoms diffusion, so that most of the Al atoms in the BM stay in ISZ-2 during the diffusion process, which makes the P-γ’ phase in this region have more favorable precipitation growth conditions. At the same time, the combined action with Ostwald ripening mechanism leads to the increase of P-γ’ phase size [32]. Figure 9 shows the SEM-EDS scanning of the joint after post-weld heat treatment, where the yellow line represents the scanning area of the SEM-EDS line. The results show that the Al content in ISZ-2 is significantly higher than in ISZ-1, which is consistent with the results of the previous analysis. On the other hand, the increase of the cubicity of the P-γ’ phase in ISZ-2 during the low-temperature aging treatment is mainly dominated by the elastic strain energy [33]. The relatively low solidus of the S-γ’ phase, indicating the S-γ’ phase is precipitated during the low-temperature aging treatment. Since no stress was applied to the BM, it is shown that the rafting of the γ’ phase is mainly controlled by temperature and time. The high temperature brazing process leads to the redistribution of the forming elements of the γ’ phase, and the solute atoms diffuse according to the chemical potential gradient. The growth of γ’ phase follows the Ostwald maturation mechanism, and the large γ’ phase engulfs the small γ’ phase. After aging treatment, the original cubic morphology was changed under the combined action of elastic strain energy and interface energy, which eventually leads to the rafting of the γ’ phase in the BM (Figure 8d).

Figure 10 shows the EBSD results of brazed joints at low magnification. It can be found that the brazed joints before and after PWHT form low angle grain boundaries (LAGBs, 2° < θ < 15°) and high angle grain boundaries (HAGBs, θ > 15°). Sheng et al. [34] found that there was composition supercooling at the front of the solid/liquid interface during isothermal solidification, and element diffusion was necessarily affected by the concentration gradient. It leads to cellular growth at the initial stage of isothermal solidification. With the solid/liquid interface moving forward, the cellular crystal nucleus deviates from the original orientation, leading to the formation of grain boundaries. Figure 10c,g show that the LAGBs of the joint after PWHT are significantly reduced, and large size grains are formed. At the same time, there is obvious stress concentration in the joint before heat treatment, and the stress concentration is significantly reduced after PWHT (Figure 10d,h). Figure 11 shows the corresponding (Geometrically Necessary Dislocation) GND density distribution charts of the brazed joints before and after PWHT. It can be found that the GND density after PWHT (1.60 × 10^15^/m^2^) is lower than the GND density before PWHT (1.66 × 10^15^/m^2^), which means that the internal stress of the joint after PWHT is reduced. All these indicates that the post-weld heat treatment process undergoes the process of recrystallization and grain growth, LAGBs merge with each other, and HAGBs swallow LAGBs, during which the internal stress in the joint can be released.

### 3.3. Effect of PWHT on Mechanical Properties of Brazed Joint

#### 3.3.1. Microhardness

Figure 12 shows the microhardness of different areas of the joint before and after the PWHT. The microhardness of the HT1 joint was in this order: NSZ > BM > ISZ-2 > ISZ-1, and the microhardness of the HT2 joint was in this order: NSZ > ISZ-1 > BM > ISZ-2. In general, the hardness value of NSZ is the highest, and the microhardness of the joint from the inside to the outside shows a tendency to decrease first and then increase. The microhardness of the HT2 joint increased overall (except for the microhardness decrease in NSZ).

In the NSZ of the HT1 joint, a large number of dense large skeleton borides are distributed, these borides have the characteristics of a hard and brittle structure, which leads to high slight hardness in NSZ. Researchers [35] have reported that the presence of boride in the joints leads to a significant increase in microhardness in this area. In HT2 joints, the microhardness of NSZ can be found to be decreased. This is mainly due to the decrease of the volume fraction of boride in NSZ of the HT2 joint, which changes from a large skeleton to a small, dispersed distribution. Therefore, it is not difficult to understand the decrease in microhardness in NSZ of the HT2 joint.

In the ISZ-1 of the HT2 joint, the morphology and size of the γ and S-γ’ phases have no obvious change compared with that of the HT1 joint, but needle boride precipitates in the ISZ-1 of the HT2 joint, which is the direct cause of the increase of ISZ-1 microhardness. The ISZ-2 of the HT2 joint was still composed of γ and γ’ phases, and no boride was precipitated. This indicates that the strengthening effect of ISZ-2 is through solid solution strengthening of the γ matrix and precipitation strengthening of the γ’ phase. The difference in volume fraction of the P-γ’ phase in ISZ-2 before and after the PWHT is not significant, but the S-γ’ phase precipitated in ISZ-2 after heat treatment, which is one of the contributors to the higher microhardness. Another contributor is the effect of γ matrix solid solution strengthening on microhardness. In other words, the strengthening effect can be considered from the perspective of solid solution strengthening elements (Re, Ru, W, Mo and Cr are all effective solid solution strengthening elements). According to the model proposed by Gypen and Deruyttere [36], the superposition of solute element strengthening can produce different effects. This can be expressed according to the following formula:(5)$$\Delta \sigma_{sol}=\sum_{i}^{}K_{i}\sqrt{C_{i}}$$
where $C_{i}$ is the concentration of solute i and $K_{i}$ is the strengthening coefficient of solute i. The solute strengthening coefficient $K_{i}$ can be obtained from the strengthening factor table of alloy elements in Ni-base alloys [37]. So far, due to the lack of relevant databases, the effect of Re on solid solution strengthening of γ matrix has not been considered. The element concentrations in different regions of ISZ before and after the PWHT of the joint were listed in Table 4. The results show that ISZ-2-$\Delta \sigma_{sol}$ is significantly larger than ISZ-1-$\Delta \sigma_{sol}$ in HT1 joints, and ISZ-2-$\Delta \sigma_{sol}$ of the HT2 joint is slightly larger than that ISZ-2-$\Delta \sigma_{sol}$ of the HT1 joint. In addition, due to the acicular boride precipitated in ISZ-1 after PWHT, the effect of γ matrix in ISZ-1 on microhardness is not considered. In conclusion, the changes of microhardness in different regions of ISZ before and after the PWHT of the joint can be explained.

The BM was mainly composed of γ and γ’ phase. After PWHT, the microhardness of BM increased, which could be attributed to γ’ phase coarsening. Some other researchers [38,39] have also found that the coarsening of γ’ phase has an adverse effect on its microhardness. 

#### 3.3.2. Tensile and Stress Rupture Properties

Figure 13 shows the ultimate tensile strength at 980 °C and the stress rupture life under the condition of 980 °C/60 MPa before and after the PWHT of the joint. The tensile strength of the HT1 joint is 409 MPa, and the tensile strength of the HT2 joint is increased to 747 MPa (Figure 13a). This is mainly related to the change of the NSZ region. the volume fraction of brittle boride in NSZ can be significantly reduced, and the distribution of brittle boride in NSZ can be changed from a large dense skeleton to a small block dispersed after the PWHT of the joint. Figure 14a,b show the local misorientation maps of the brazed joint, indicating that the HT1 joint has significant stress concentration at the interface around the bulk boride compared with the HT2 joint. Under the action of external stress, cracks originate preferentially generated around the boride and then propagate along the boride (Figure 14c). Continuous bulk brittle borides exhibit low resistance to crack growth. When the borides are discontinuous, the crack propagates to γ solid solution, which will encounter a large resistance to prevent the crack from further propagating. In other words, it is necessary to apply more external force to the joint in order to propagate the crack further until the final fracture of the joint, as shown in Figure 14d. Therefore, the tensile strength of the joint after PWHT is significantly improved.

The stress rupture life (967 h) of the HT2 joint is significantly longer than that of the HT1 joint (268 h), as shown in Figure 13b. This may be related to the increase of the volume fraction of the S-γ’ phase and the migration of more solid solution strengthening elements to the center of the joint after a long time of aging treatment, which enhanced the solid solution strengthening mechanism of the joint. In particular, the characteristic elements Re and Ru in the fourth-generation nickel-based single crystal superalloy enter the joint, which can reduce the stacking fault energy and increase the lattice misfit between the γ and γ’ phase, respectively [40,41,42]. It is generally believed that grain boundaries are considered a crystal defect at sustained high temperatures, which in turn affects the high-temperature creep properties of the joints [43]. It can be seen from Figure 14e,f that the joint fracture mostly breaks along the grain boundary. Furthermore, the number of HAGBs is increased in the joint of HT2 compared to the joint of HT1 (Figure 10), indicating that the crack initiation is mainly at the HAGBs. The crack propagation along the interface between HAGBs and borides was observed from the fracture path of the joint (Figure 14f). It is worth noting that the MC carbide was found to change from large blocks to small pieces of particles after the PWHT of the joint and was uniformly distributed on the HAGBs, as shown in Figure 7. This suggests that MC carbide can strengthen the grain boundary and prevent the rapid failure of the joint due to the grain boundary defect.

Figure 15 shows the fracture morphologies of brazed joints before and after the PWHT after tensile and stress rupture tests. All the samples were fractured at the center of the joint, and the fracture surface is relatively flat without plastic deformation. The tensile fracture surface shows obvious quasi-cleavage fracture characteristics, and the river pattern and tear edge can be observed obviously. Many pores were observed on the fracture surface of the brazed joint without the PWHT (Figure 15a), which was unfavorable to the mechanical properties of the brazed joint. However, the surface morphology of stress rupture fracture was obviously different from that of tensile fracture. A large number of small creep dimples were observed, showing the characteristics of intergranular fracture of microporous aggregation. Furthermore, the number of fine dimples in the fracture surface of the joint after the PWHT was significantly increased, and no holes were observed on the fracture surface of the joint (Figure 15d), and the stress rupture life of the joint was improved.

## 4. Conclusions

The microstructure and mechanical properties of brazed joints of the fourth-generation nickel-based single crystal superalloy before and after the PWHT were investigated. The following conclusions could be drawn:The non-isothermal solidification zone (NSZ) and the isothermal solidification zone (ISZ) were observed in the brazed joint. The NSZ was composed of M_3_B_2_, MB-type boride, MC carbide and γ matrix, and the ISZ was consisted of γ and γ’ phases with different sizes. The simulation results of the residual liquid phase in the joint are basically consistent with the experimental results.After the PWHT, the boride in NSZ changed into discrete block distribution and needle boride precipitated in ISZ-1. The formation and evolution of P-γ’ in ISZ-2 and S-γ’ in ISZ-1 are mainly related to the diffusion effect of boride on Al and Ta atoms. In addition, high-angle grain boundaries formed in the joint, which is related to the nucleation and growth of grains during recrystallization.The microhardness of the joint (except NSZ) after the PWHT is slightly higher than that before the PWHT. The boride distribution controls the microhardness of NSZ, and the microhardness of ISZ is affected by the solid solution strengthening of the γ matrix and precipitation strengthening of the γ’ phase.After the PWHT, the tensile strength and stress rupture life of the joint are significantly improved. MC carbides pinned at grain boundaries and the discontinuous blocky borides play a critical role in the mechanical properties of the joint. The fracture surfaces of tensile and stress rupture joints are quasi-cleavage fracture and intergranular fracture of microporous aggregation, respectively.

## Figures and Tables

**Figure 1 materials-16-03008-f001:**
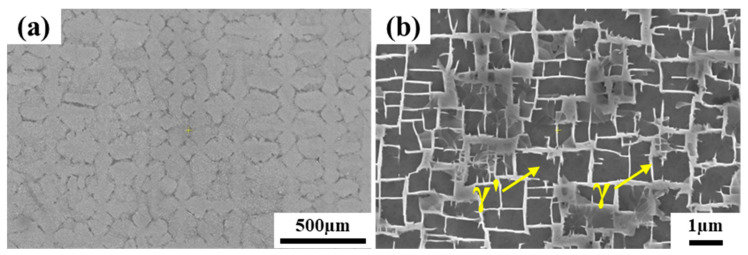
Microstructure of as-cast single crystal rod: (**a**) Dendrite morphology, (**b**) γ/γ’ phase.

**Figure 2 materials-16-03008-f002:**
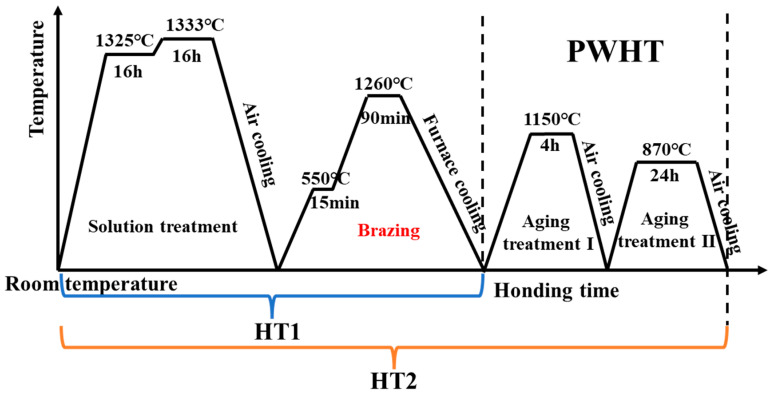
Schematic representation of the time-temperature relationship between the brazing process and PWHT.

**Figure 3 materials-16-03008-f003:**
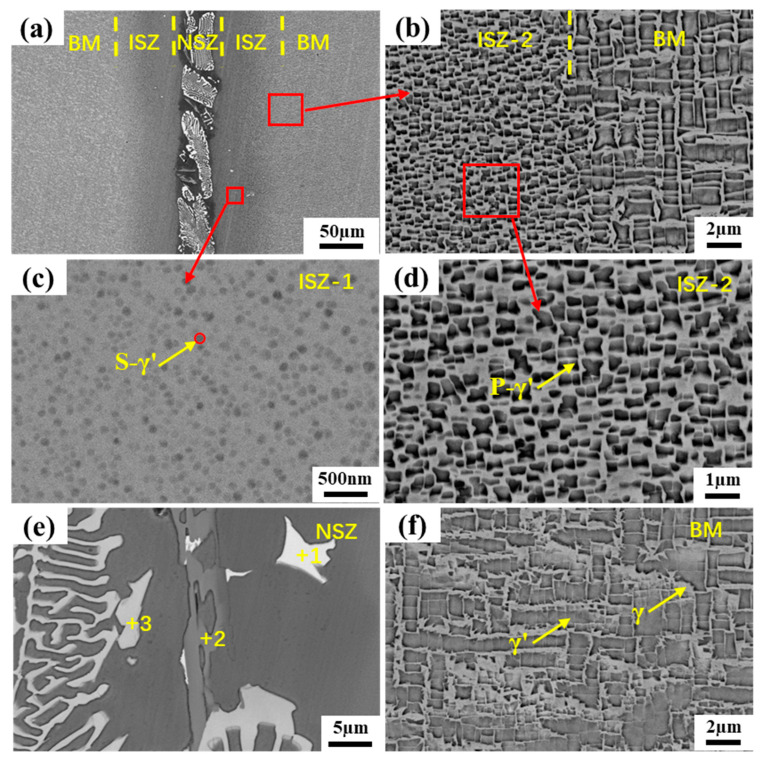
Microstructure of the joint brazed at 1260 °C for 90 min: (**a**) SEM images of the whole joint, (**b**) Microstructure of between ISZ-2 and BM, (**c**) Microstructure of ISZ-1, (**d**) Microstructure of ISZ-2, (**e**) Microstructure of NSZ and (**f**) Microstructure of BM.

**Figure 4 materials-16-03008-f004:**
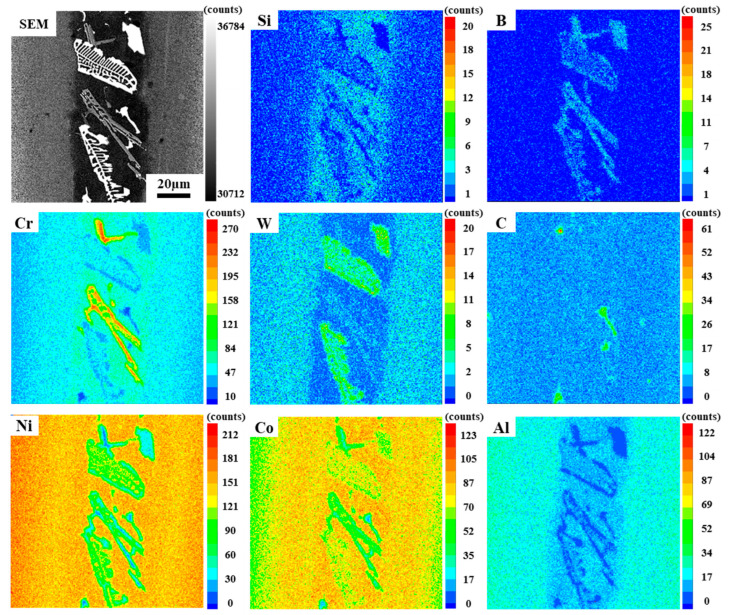
The element distribution of Ni, Co, Cr, Al, C, W, B and Si in the brazed joint.

**Figure 5 materials-16-03008-f005:**
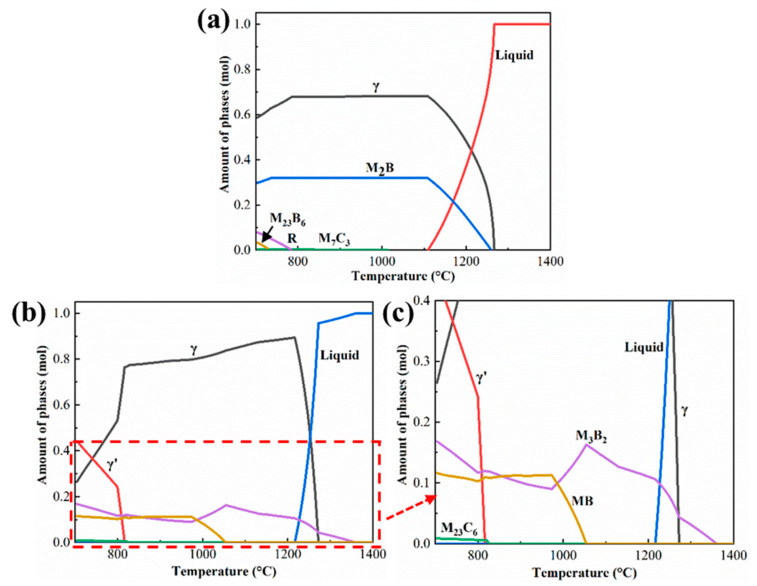
The solidification process of the brazed joint simulated by Thermal-calc 2021a software: (**a**) The solidification process of filler alloy without considering the dissolution of BM, (**b**) the solidification process of filler alloy considering dissolution of BM, and the (**c**) enlargement of partial area in (**b**).

**Figure 6 materials-16-03008-f006:**
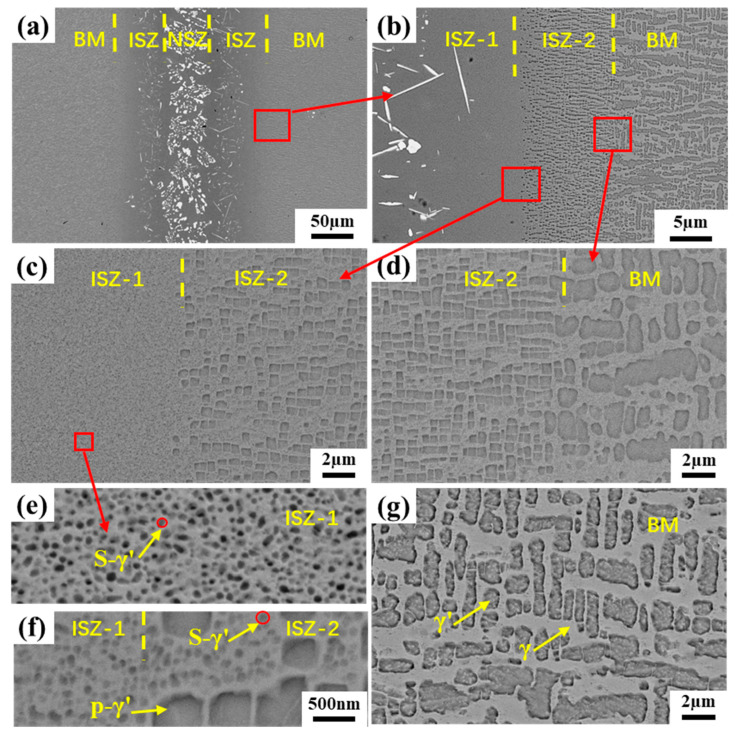
Microstructure of joint after the PWHT: (**a**) SEM images of the whole joint, (**b**) enlarge the image between ISZ and BM, (**c**) microstructure of between ISZ-1 and ISZ-2, (**d**) microstructure of ISZ-2 and BM, (**e**) microstructure of ISZ-1, (**f**) the S-γ’ phase in the γ matrix channel in ISZ-2 and (**g**) microstructure of BM.

**Figure 7 materials-16-03008-f007:**
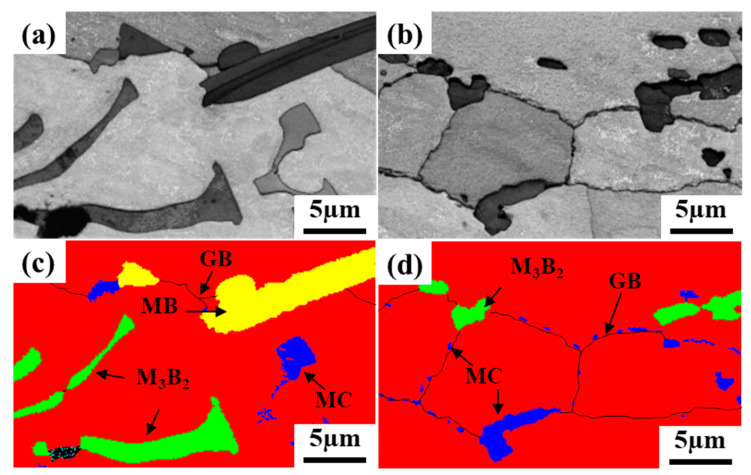
EBSD results of brazed joints before and after PWHT: (**a**,**b**) band contrast microstructural images, (**c**,**d**) phase distribution map.

**Figure 8 materials-16-03008-f008:**
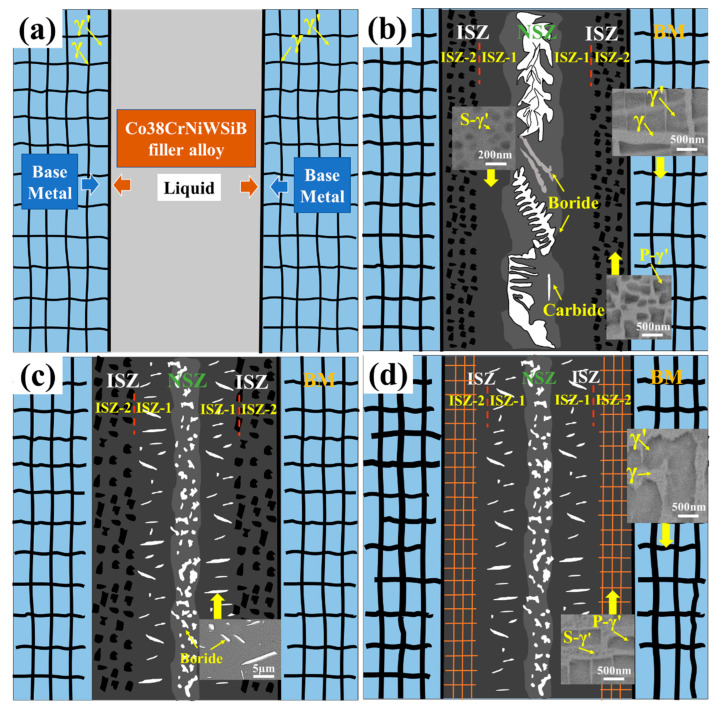
Schematic diagram of brazed joint formation and microstructural evolution after the PWHT: (**a**) interdiffusion of filler metal and BM, (**b**) completion of solidification of residual liquid, (**c**) high-temperature aging treatment stage, and (**d**) low-temperature aging treatment stage.

**Figure 9 materials-16-03008-f009:**
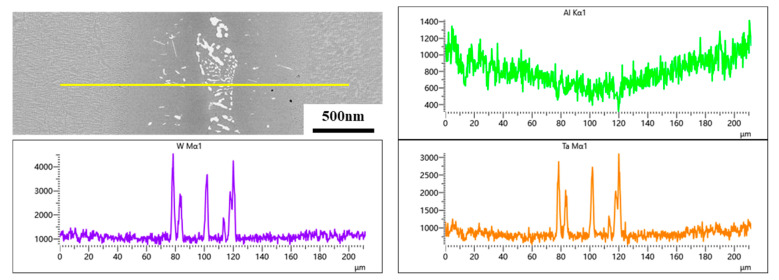
The SEM-EDS line scan results of joint after the PWHT.

**Figure 10 materials-16-03008-f010:**
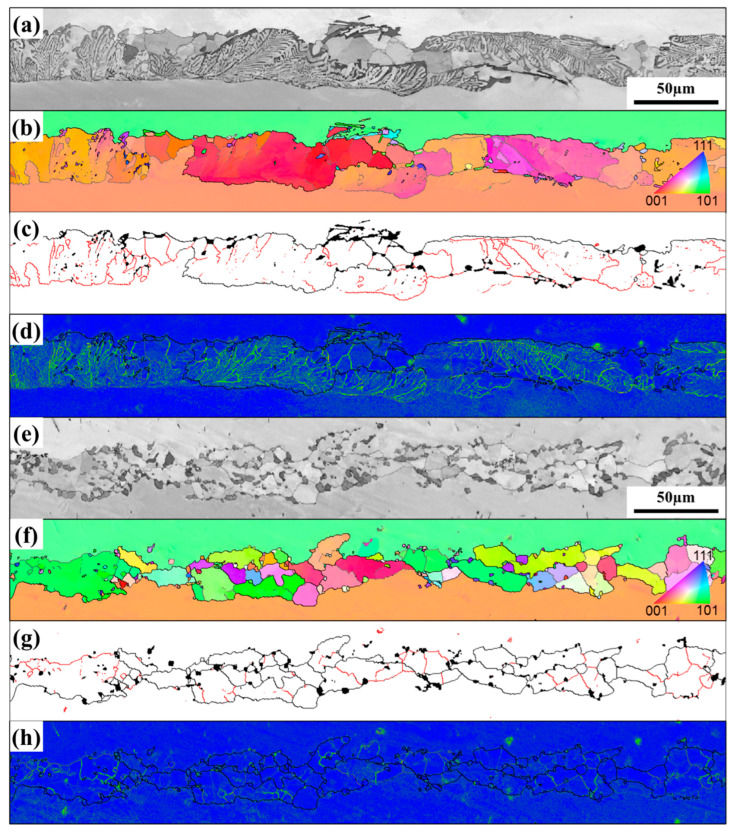
EBSD results of brazed joints before and after PWHT: (**a**,**e**) band contrast microstructural images, (**b**,**f**) the IPF maps, (**c**,**g**) grain-boundary maps (Red line denotes grain-boundary misorientation between 2°and 15°, while black line denotes grain-boundary misorientation larger than 15°. In addition, (**d**,**h**) show the kernel average misorientation (KAM) maps.

**Figure 11 materials-16-03008-f011:**
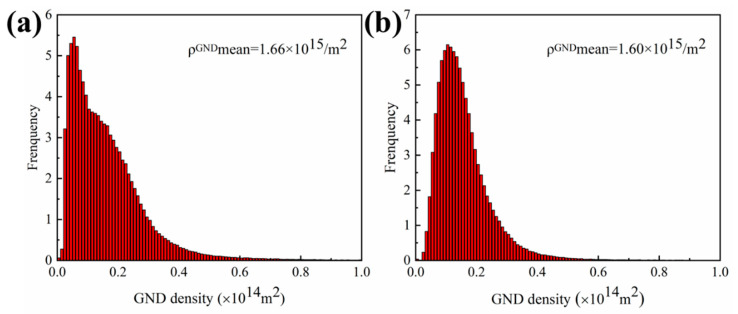
GND density of the brazed joint: (**a**) before PWHT, (**b**) after PWHT.

**Figure 12 materials-16-03008-f012:**
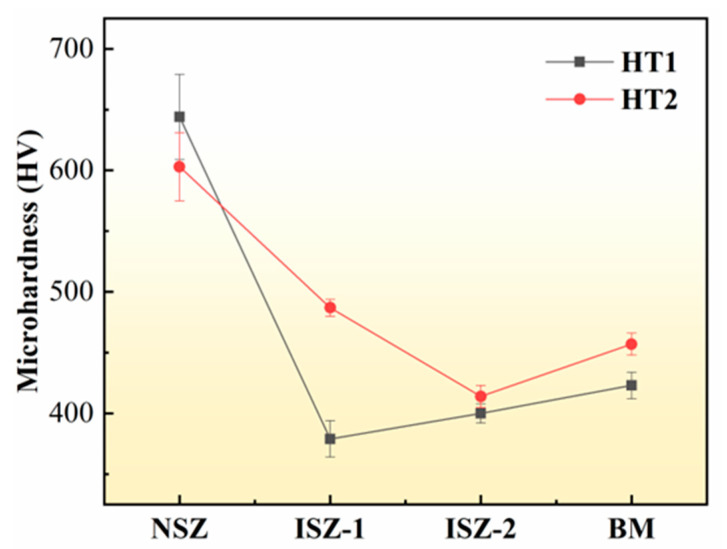
Microhardness change of NSZ, ISZ-1, ISZ-2 and BM of the HT1 joint and the HT2 joint.

**Figure 13 materials-16-03008-f013:**
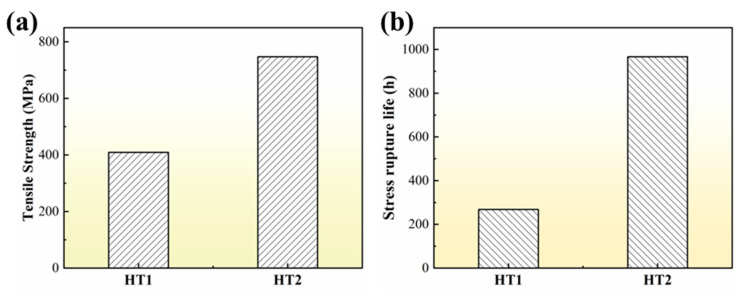
Mechanical properties of brazed joints before and after the PWHT: (**a**) high temperature tensile, and (**b**) stress rupture life.

**Figure 14 materials-16-03008-f014:**
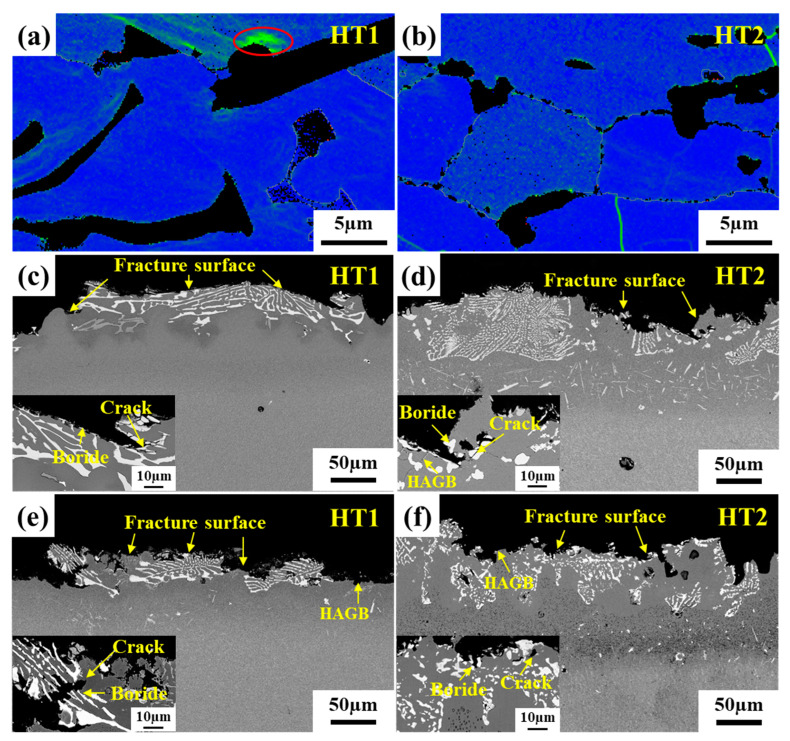
Brazed joints before and after PWHT: (**a**,**b**) KAM maps of brazed joint, (**c**,**d**) fracture surface of high temperature tensile specimens, (**e**,**f**) fracture surface of stress rupture specimens.

**Figure 15 materials-16-03008-f015:**
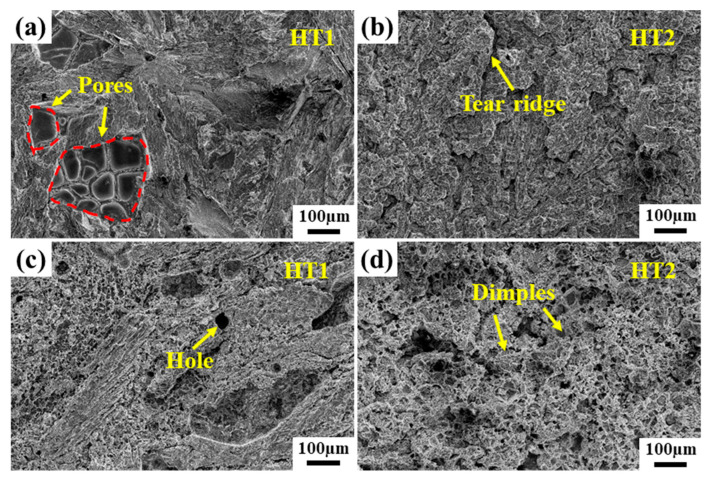
Fracture morphology of brazed joints before and after the PWHT: (**a**,**b**) high temperature tensile (**c**,**d**) stress rupture.

**Table 1 materials-16-03008-t001:** Chemical composition (wt.%) of base metal alloy.

Element	Co	Al	Cr + Mo + Ta + Hf	W	Re	Ru	Ni
Content	10~14	4~6	9.5~16.6	5~7	4~7	3~5	Bal.

**Table 2 materials-16-03008-t002:** Chemical composition (wt.%) of Co38CrNiWSiB filler alloy.

Element	Co	Cr	Fe + La + Mn + W	B	Si	C	Ni
Content	38	21.8	14.58	2.04	1.98	0.02	21.6

**Table 3 materials-16-03008-t003:** EDS results of chemical composition (at.%) at point 1–3 in Figure 3e.

Position	Co	Ni	Cr	C	Ta	Re	Ru	W	Hf
1	1.65	1.96	3.52	66.70	23.23	-	-	0.97	1.97
2	11.55	3.03	70.92	-	-	6.34	4.83	2.87	-
3	30.91	19.11	14.76	-	11.75	-	-	23.48	-

**Table 4 materials-16-03008-t004:** EDS results showing the compositions of the ISZ-1 and ISZ-2 (at%).

Elements		Cr	Ru	Re	Ta	Al	Co	W	Ni
HT1	ISZ-1	13.8	0.73	0.88	1.22	8.2	27.5	3.6	Bal.
	ISZ-2	7.33	1.45	2.25	2.94	11.14	15.48	2.62	Bal.
HT2	ISZ-1	13.38	1.0	1.22	1.32	6.82	26.16	4.2	Bal.
	ISZ-2	7.49	1.67	2.08	2.70	12.18	16.41	3.17	Bal.

## Data Availability

The data used to support the findings of this study are included within the article.

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
