# Peer review of "Effects of Post-Weld Heat Treatment on Microstructure and Mechanical Properties of the Brazed Joint of a Novel Fourth-Generation Nickel-Based Single Crystal Superalloy"

_materials, 2023, doi:10.3390/ma16083008_

Round 1

Reviewer 1 Report

This is a good paper on the effect of PWHT on the microstructure and mechanical properties of a brazed SX superalloy. I recommend this paper for publication in Materials after some minor revisions:

  1. Author mentioned that the brazing experiment after solution treatment will not lead to the remelting of the filler alloy. Did you mean remleting of the base metal? Please clarify.
  2. Please clarify the gamma prime rafting mechanism in the base metal adjacent to the bonding zone. What was the effect of PWHT on the rafting? Please elaborate.
  3. If you are using a justification style (e.g. calculation of solid solution contributions to the joint hardness/strength) which is used in a previous work to analyze your data, it is more eligible to mention directly and clearly who first used this type of analysis.

Reviewer 2 Report

Manuscript titled "Effects of post-weld heat treatment on microstructure and mechanical properties of the brazed joint of a novel fourth-generation nickel-based single crystal superalloy" is a nice work related to a new development PWHT of a nickel-based superalloy.

Recommendation: Minor revision

Before publishing, the authors must revise the paper as per comments mentioned below:

1.     Copyright permissions for figures taken from published papers should be checked. Plagiarism must be below the limit prescribed by the journal. This is just a general reminder comment only. Reviewer has not checked the plagiarism or copyrights.

2.     Research gaps and objectives have been presented well in this paper.

3.     Experimental procedures and results have been discussed well.

4.     Overall, the paper is of good quality but needs some minor improvements before publishing.

5.     Although the simulation results of the residual liquid phase in the joint are consistent with the experimental results. But still if it is possible to somehow quantify the mismatch?

6.     In addition, formation of high angle grain boundaries in joints during PWHT due to nucleation and growth of grains during recrystallization due to stress concentration. This sentence needs to be revised properly.

7.     English grammar to be thoroughly checked and corrections to be made especially in Abstract and Conclusion sections.

Reviewer 3 Report

The subject of research presented in the paper is effects of post-weld heat treatment on microstructure and mechanical properties of the brazed joint of a novel fourth-generation nickel-based single crystal superalloy. Authors of this article brazed the fourth-generation nickel-based single crystal superalloy. Brazed was carried out vacuum with a Co-based filler alloy. The microstructure and mechanical properties of as-brazed and heat-treated joints were studied. The formation of precipitates in the joints and the microstructure evolution of the joints during heat treatment were discussed.

In the article presented for review, the authors described their research very meticulously and with great care. All the results contained in it were presented in a very clear and legible way for the recipient. The conclusions and the study summary are consistent and well-formulated. Summing up, the reviewed work presents a very high substantive and experimental value.
